# Delphi consensus study to develop guidelines for the management of adults with borderline personality disorder in the emergency department: a protocol

Aaron Prosser [1], Victor Hong,[2] Bartosz Helfer [3,4] David Fudge,[1] Janet Patterson,[1] Patricia Rosebush,[1] Benicio N Frey [1,5,6] Paul Links[1]

For numbered affiliations see end of article.

**Correspondence to**
Dr Aaron Prosser;
aaron.prosser@medportal.ca

## ABSTRACT

**Introduction** Clinicians caring for adults with borderline personality disorder (BPD) in acute settings such as the emergency department (ED) have little evidence/guidance to base decisions on. Specific/detailed guidance for managing BPD in the ED is needed given the morbidity and mortality risks, high service utilisation, unique challenges and risk of iatrogenic interventions. The primary objective of this study is to use a consensus method to develop a guideline for managing adults with BPD in the ED. This protocol and the key questions for the guideline were developed with the advice of people with BPD and their family members/support persons.

**Methods and analysis** We will perform a four-phase Delphi study of an expert panel of clinicians, researchers, adults with BPD and their family members/support persons. Various disciplines (psychiatry, psychology, emergency medicine, nursing, social work) and treatment approaches will be included in the expert panel. An online questionnaire will be developed from systematic reviews, qualitative assessments of pivotal literature, and opinions suggested by the panel (phase 1). The panel will rate their agreement on opinions for each key question covering areas of emergency care of adults with BPD using two rounds of this questionnaire (phases 2 and 3). Opinions meeting predefined thresholds for consensus will be brought to consensus meetings moderated by an independent chair (phase 4). The purpose of these meetings is to finalise the set and phrasing of the opinions for each area of emergency care. These final opinions will be the recommendations in the guideline. If there are significant differences of opinion, the guideline will present both recommendations alongside one another.

**Ethics and dissemination** This study has received ethics approval by the Hamilton Integrated Research Ethics Board in Hamilton, Ontario, Canada. The results of this study will be disseminated through peer-reviewed publications, conferences and national professional and patient/family/support associations.

## INTRODUCTION
### The need for guidelines

Borderline personality disorder (BPD) is a highly heritable (~55%) condition whose core pathology is a disturbance in identity, attachment and emotion regulation.[1 2] BPD is often viewed as a trauma disorder since for many it emerges from psychosocial adversity interacting with a genetic diathesis.[1 2] The symptomatic expression of BPD includes dysphoria, dysregulation, recurrent self-harm/suicide attempts and conflicted/unstable relationships.[3] These symptoms are often severe and disabling, causing frequent crises and presentations to the emergency department (ED), with frequent suicidal thoughts and self-harm behaviours being behavioural trademarks of the disorder. Specialised outpatient treatments for BPD have been developed and shown to be effective in randomised controlled trials (RCTs).[4] There is also limited RCT evidence for some medications targeting specific clusters of

symptoms (eg, emotion dysregulation, impulsivity).[5] [6] These developments are a major accomplishment and have informed guidelines on BPD, notably the National Institute for Health and Care Excellence (NICE),[7] American Psychiatric Association (APA)[8] and National Health and Medical Research Council (NHMRC)[9] guidelines.

## STRENGTHS AND LIMITATIONS OF THIS STUDY

⇒ The Delphi method will be used due to the limited research on managing adults with borderline personality disorder (BPD) in the emergency department to inform a guideline.

⇒ Our protocol incorporated the advice and feedback of people with BPD and their family members/support persons.

⇒ Our methods use a questionnaire to minimise the disadvantages of group discussion (eg, domination by individuals or pressure to conform) while the consensus meeting capitalises on its advantages (eg, clarifying ambiguities or editing the phrasing of recommendations).

⇒ The expert panel will include participants from different disciplines, backgrounds and treatment approaches including adults with BPD, their family members and support persons, clinicians and researchers.

⇒ The expert panel might not be fully representative since recruitment is not based on random sampling and the recommendations will be based on consensus, which is based on shared knowledge and practical experience rather than scientific proof.

While crises and acute treatment are addressed in these guidelines, the sections are brief since most of the evidence pertains to outpatient settings. Therefore, we formed a Guideline Working Group (GWG) to oversee the development of a guideline on managing BPD in the ED. Our focus is on adults with BPD, although certainly many of the principles may apply to adolescents. Our primary aim will be to identify a set of recommendations for each key question (KQ) using Delphi consensus methods. Each KQ defines a key area of the emergency care of adults with BPD. Specific and detailed guidance for managing adults BPD in the ED is needed given the common pitfalls related to high service utilisation, unique challenges in the emergency setting, risk of iatrogenic interventions and recurrent suicide attempts and self-harm.

### High service utilisation and unique challenges

Adults with BPD are frequent visitors to the ED, representing upwards of 9% of all visits in some jurisdictions, with recurrent visits being common.[10] Reasons for presenting to the ED are numerous, including self-harm, suicide attempts, situational crises, adjustment reactions, depression, anxiety, emotion dysregulation, agitation, intoxication, withdrawal, substance induced psychosis and housing/financial problems. There are patient, staff and systems factors which create unique challenges to managing BPD in ED settings.[11] Some are listed in table 1.

### High risk of iatrogenic interventions in the emergency department

Adults with BPD appear to be at higher risk of iatrogenic interventions in all clinical settings, particularly acute settings such as the ED.[11] These include inadequate risk assessment, unnecessary inpatient hospitalisations, excessive use of medications or unnecessary use of seclusion or physical/chemical restraints. Staff attitudes and countertransference reactions can also drive iatrogenesis. Staff who are otherwise empathic, validating, supportive and effective clinicians may have negative or positive reactions which—if not examined, accepted and contained—can be the driver of harmful interventions.[12] Negative

reactions include withdrawal, rejection, hostility, helplessness, hopelessness, anxiety or feeling overwhelmed. Positive yet often unhelpful reactions include rescue fantasies or feeling excessively protective of or responsible for a patient. Lack of awareness of the separation of acute and chronic risk factors may play a role in these reactions. Given the prevalence of comorbidities in individuals with BPD, this adds further complexity to the clinical decision making.[13] Iatrogenesis is likely caused by an interplay between these patient, staff and systems factors, often amplified in the ED.

### Risk of recurrent suicide attempts and self-harm

Recurrent suicidal acts, threats, urges and self-harm behaviours are defining symptoms of BPD.[3] The risk of suicide is 3%–10% in people with BPD and the average number of suicide attempts per individual is 3, with completed suicide occurring about once per 23 attempts.[1] About 75% of people with BPD self-harm and among these 90% repeatedly self-harm.[1] Taking on this risk is inherent to the work of caring for people with BPD. These risks are greatest for clinicians working in the ED because people with BPD presenting to the ED often do so because they are acutely dysphoric, actively suicidal, engaged in severe self-harm and/or attempted to end their life. However, discerning the difference between acute versus chronic suicide/self-harm risk is difficult when a patient is chronically at high risk. This is especially the case when the risk assessment/management must be done by a clinician who does not have a longitudinal relationship with the patient, which is the reality for most clinicians working in the ED. For these reasons, many clinicians experience fear not only for the safety of the people with BPD they care for, but also their own medicolegal liability. This is yet another reason why specific and detailed guidance for managing BPD in the ED is needed, especially regarding suicide/self-harm risk assessment/management.

## METHODS AND ANALYSIS
### Rationale for the Delphi method

Our project started by first determining the most appropriate method to develop the guideline. Guidelines are usually constructed from systematic reviews and meta-analyses of primary studies. However, our group was

**Table 1** Challenges to managing BPD in ED settings

| Patient | Staff | System |
|---|---|---|
| ► Recurrent ED visits and high service utilisation.<br>► Disruptive behaviours.<br>► Chronic high risk for suicide and self-harm.<br>► Transference reactions. | ► Problematic staff attitudes.<br>► Varying levels of training.<br>► Ill-defined/time-limited responsibility for patients.<br>► Countertransference reactions. | ► Transient/changing ED staff.<br>► Changing ED workloads.<br>► Disposition problems.<br>► Availability of inpatient beds, aftercare services and rapid outpatient follow-up. |

BPD, borderline personality disorder; ED, emergency department.

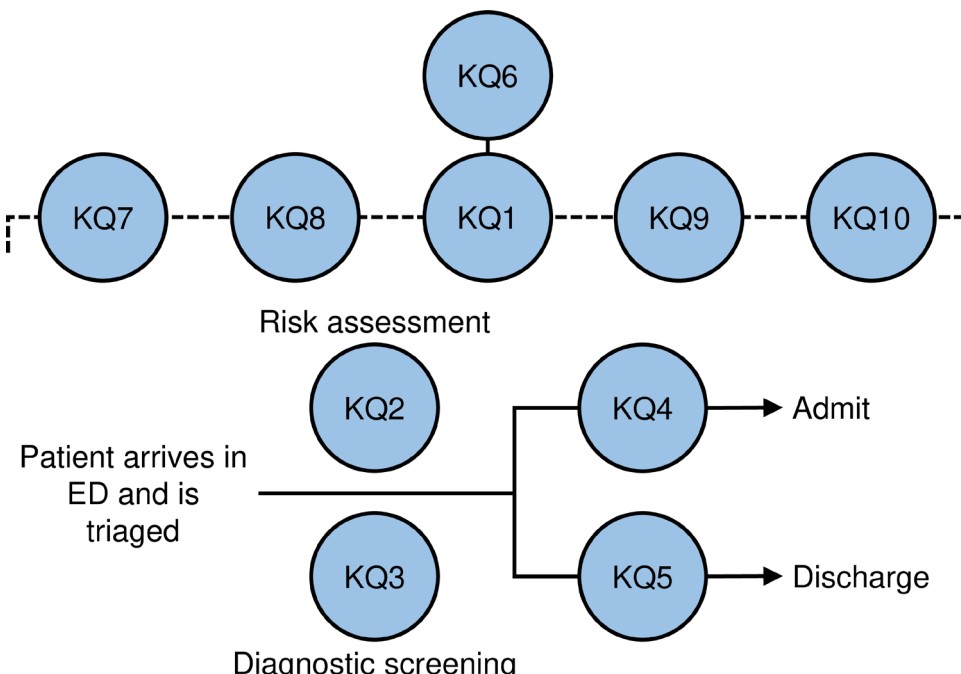

**Figure 1** Analytical framework and key questions (KQs). ED, emergency department.

already aware that there is scant research in this area. This was confirmed by pilot systematic searches. Therefore, we decided to conduct a Delphi consensus study to come up with recommendations from clinicians, researchers, patients and family members/support persons.

### Development of KQs

Our KQs were built from the two main tasks which need to be addressed for all people with BPD who come to the ED. We used the analytical-graphical model as developed by Nelson *et al*[14] to visualise the analytic framework and our KQs (figure 1). A primary consideration in the ED is the risk assessment (KQ2) and, if a diagnosis is not established yet, screening for BPD (KQ3). This usually starts with the emergency medicine team who may or may not ask the psychiatrist and/or mental health clinicians (eg, psychiatric nurses, social workers) for further assistance in the assessment and disposition planning. Another consideration is to decide if to admit the patient to hospital (KQ4) or discharge the person to aftercare psychosocial services (KQ5). Alongside this process is the overall care of the person with BPD while they are in the ED (KQ1). Here we mean the general interventions to use with all patients throughout their stay in the ED (eg, empathy, validation, management of countertransference and bias) and specific interventions (eg, psychoeducation, medications) to address the specific needs of a patient. A related key area are interventions for the management of agitation and aggression in people with BPD in the ED (KQ6). Another key area is the involvement and role of family members and support persons (KQ7) and outpatient providers (KQ8) while people with BPD are in the ED. The last key areas are approaches to providing optimal medical care (KQ9) and effective environmental

elements (eg, physical layout, interior design) for people with BPD presenting to the ED in a mental health crisis (KQ10).

Based on this analytical framework, we constructed 10 KQs:

1. What psychosocial and pharmacological interventions in the ED are associated with benefit while minimising harm for adults with BPD, with considerations regarding potential comorbidities?
   a. What interventions in the ED are effective?
   b. What interventions in the ED are harmful?
2. What are the effective approaches to assess and manage the risk of suicide and self-harm of adults with BPD, with considerations regarding potential comorbidities?
3. What should be included as part of the emergency evaluation of adults to screen for previously undiagnosed BPD?
4. What is the role of hospitalisation in managing adults with BPD?
   a. What are the indications for hospitalisation?
   b. What is the effectiveness of hospitalisation?
   c. What are the harms of hospitalisation?
5. What psychosocial aftercare services (including partial hospitalisation) can help adults with BPD who present to the ED but are not hospitalised?
6. What interventions in the ED for agitation and aggression are associated with benefit while minimising harm for adults with BPD?
   a. What interventions for agitation and aggression are effective?
   b. What interventions for agitation and aggression are harmful?

7. What are effective approaches to involving family members and support persons in the care of adults with BPD in the ED?

8. What are effective approaches to involving outpatient providers in the care of adults with BPD in the ED?

9. What are effective approaches to providing optimal medical care for adults with BPD presenting in a mental health crisis to the ED ?

10. What are the most effective environmental elements (physical layout and interior design) for adults with BPD in the ED?

We will systematically review the literature on each KQ, the results of which will be integrated into the Delphi process (see methods section for details). The Delphi process will be used to provide guidance on each of these KQs. It will draw on the expertise of people across disciplines (psychiatry, psychology, emergency medicine, nursing, social work), backgrounds (clinicians, researchers, patients, family, support persons) and treatment approaches. The latter will hopefully include Dialectical Behaviour Therapy (DBT), Transference-Focused Psychotherapy (TFP), Mentalisation-Based Treatment (MBT), Good Psychiatric Management (GPM), Schema Focused Therapy (SFT), Cognitive–Behavioral Therapy (CBT), Systems Training for Emotional Predictability and Problem Solving (STEPSS), Integrated Modular Treatment (IMT) and psychopharmacology. Other content experts who may not align with a specific treatment approach will be contacted for participation.

### Patient and public involvement

While developing this protocol and the KQs included here, we sought advice from people with BPD and their family members/support persons on what are the most important questions to include from their perspective as well as feedback on an earlier draft of the protocol. VH contacted potential participants by using the listserve from several advocacy organisations, and 13 provided constructive feedback including 3 persons with BPD and 9 family members and 1 support person of someone with BPD. Participants were asked to review the KQs, provide feedback and propose other KQs. All 13 participants were those who themselves or had supported a person who had experience with an emergency department presentation for BPD-related issues. They agreed on the set of proposed KQs and four participants included specific suggestions. One person with BPD wanted KQ8 to be explored further. Family members and the support person requested additional KQs and edits, all of them were applied after consultation with the GWG and are listed here: (1) two additional KQ (9 and 10) were added, (2) the order of questions was changed (KQ4 became KQ1 and the numbering of the others was adjusted accordingly) and (3) 'support person' was added to KQ7 and the word 'aggression' to KQ6. We have also edited the methods of the guideline development process below to include the wider support circle of people with BPD (going beyond family members or caregivers). Ten others expressed interest in participating

in this phase of the project. One did not meet criteria because they did not have BPD, one ultimately did not choose to participate and eight did not respond to the protocol when it was shared. Additional feedback included preferences relating to the need to consider biases based on socialeconomic status, race and gender as well during the guideline development process. Their preferences will be included in the Delphi process and shared with the guideline development team. Moreover, special attention will be given to these biases while preparing the systematic review. The guidelines will be shared with the participants in the development of guidelines.

### Study design

This study will use the Delphi method in order to achieve consensus across the group of participants. This method is appropriate for our purposes because it maximises the benefits of group discussion while minimising its disadvantages.[15] The main disadvantages are domination by powerful individuals, coalitions, vested interests, the biasing effects of personality and seniority, that only one person can speak at a time, and group pressures for conformity. The Delphi method provides a structured, democratic, transparent and iterative procedure with controlled feedback to participants to determine the extent of agreement among a group about a given issue.[15 16] The procedure will consist of questionnaire development (phase 1), rating evaluation (phase 2), rerating (phase 3) and consensus meeting (phase 4). Phase 1 will include systematic reviews of our KQs to incorporate any recommendations which may come from the existing literature.

### Study governance

The GWG is composed of six psychiatrists (VH, DF, JP, PR, BNF and PL), a psychiatry resident (AP) and a meta-researcher (BH) with expertise in guideline development. All clinicians in the GWG currently treat people with BPD in outpatient, inpatient and emergency settings. One of the members (PL) was a codeveloper of GPM[1 17] and another (VH) published an expert opinion on the management of BPD in the ED based on GPM principles.[11] As a result, PL and VH will also be participants given their recognised expertise. However, this also is a potential source of bias. We detail how we will mitigate this (online supplemental appendix A). Study methods for the protocol were established in virtual meetings and email correspondence. The GWG will meet at minimum once quarterly and at higher frequency as needed until projected completion. Meetings will be held virtually and we will have regular email correspondence. To further address any outstanding issues with study governance, we will follow the complete interaction approach with full transparency between the systematic review team and eventual guideline developers.[18]

### Involvement of persons with BPD and their families/support persons

Adults with BPD, their family members and support persons will have a voice in these guidelines since the

recommendations will directly impact them. We recognise that there may be differences of opinion between patients/family/support persons and clinicians/researchers on some questions during the Delphi process. If there are significant differences in opinion, we will be transparent about these differences in the guideline. This ensures that patient/family/support persons voices are not silenced by majority opinion since we anticipate there will be more clinician/researcher participants. Patients or patients' representatives will be continuously involved throughout the guideline development process following a modified model by Armstrong *et al.*[19] Following NICE recommendations, at least two patient/caregiver/advocate members will be involved in the Delphi process and consensus meeting as panellists.[20]

### Participant selection/recruitment

Participants will be included if they are clinicians and/or researchers who have significant experience in the management of BPD in the ED. We will also recruit patients and family members/support persons. The clinician/researcher sample will be recruited using a number of strategies. First, we will contact national professional organisations for emergency medicine, psychiatry, nursing and social work in English-speaking countries, specifically the USA, Canada, the UK, Ireland, South Africa, Australia and New Zealand. We will ask them to advertise our study through their email mailing lists. Second, the GWG is already aware of a number of experts who we will invite to participate. These individuals were either a developer or published expert in a treatment approach for BPD, a clinician/group member on the NICE, APA and NHMRC guidelines, or is a published expert in the management of BPD in the ED. Third, the publication of this protocol will also help advertise this study. Patients and family members/support persons will be recruited by contacting national patient advocacy and support organisations. We will ask these organisations to advertise our study through their email mailing lists and social media to help with recruitment. We will ask all participants to provide suggestions too. Readers of this protocol may also give suggestions. Suggestions can be made to the GWG by contacting the corresponding author of this protocol (aaron.prosser@medportal.ca). The deadline for suggesting a participant is 3 months after the publication date of this protocol.

### Representation

We believe that representation is important to the guideline development process. Thus, we will attempt to build a balanced panel of experts based on the following demographic data: discipline (psychiatry, psychology, nursing, emergency medicine, social work), group (clinician, researcher, adult with BPD, family member/support person), treatment modality allegiance, ethnicity, gender and sexual orientation. In order to do this, at the recruitment stage, the study coordinator will email a form to potential participants who have expressed interest to

participate. In this form, participants will be invited to provide basic demographic data to allow us to build a balanced panel. There will be an option to select 'prefer not to answer', 'do not know' and 'other' for all questions. If a demographic factor has reached saturation, the person may be ineligible to participate. This demographic data will be collected before consent is obtained to participate in this study. Therefore, if a person is ineligible to participate or does not consent to participate in the study after reviewing the consent form, the demographic form will be promptly deleted and will not be kept part of the study records. If the person is eligible to participate and consents, the form will be retained as part of the study records, deidentified and statistically summarised so that we can show the representativeness of the expert panel in the publication of the guideline. Saturation will be determined on a case-by-case basis depending on our pool of potential participants and the representativeness of those who have already consented to participate.

### Delphi procedure

The Delphi process is an iterative procedure which typically proceeds in three phases.[16] We will add a fourth phase in order to have a consensus meeting moderated by an independent chair, thus combining the Delphi method with the consensus conference method. This is because we believe that an open-ended group discussion has advantages over questionnaires. In particular, group discussion can be helpful for clarifying ambiguities and differences and editing phrasing for the final recommendations of the guideline. Furthermore, a meeting provides an opportunity to reach consensus on issues where consensus has not yet been achieved. Questionnaires will be delivered and collected from participants using REDCap. Data will be deidentified by giving participants an arbitrary study ID linked to their questionnaire responses, however, no identifying data will be collected (eg, names, age, date of birth) with the questionnaire responses.

### Phase 1: questionnaire development

Phase 1 has three goals: (1) create the expert panel, (2) create a pool of opinions on each KQ from three sources of information and (3) create an online questionnaire using this pool of opinions for rating evaluation by the expert panel. The expert panel will be created using the selection and recruitment strategies described above.

The first source of information we will use for generating the opinion pool will be systematic reviews of the literature on each KQ. Full details of our methods, including search syntax, are in online supplemental appendix B. Summary opinions will be generated by the GWG based on the results of these systematic reviews.

The second source of information is a qualitative assessment of pivotal guidelines, psychotherapy manuals, book chapters and articles (online supplemental appendix C). This allows us to include pivotal literature which will not necessarily be identified in the systematic reviews (eg,

psychotherapy manuals). AP will search these sources to extract a narrative summary of the opinions on each KQ. Quality control will be confirmed by an independent assessor on a random 20% subset of the opinions from these pivotal sources.

The third source of information will be our expert panel. We will invite the expert panellists to provide written opinions on each KQ once the participant has consented to participate. They will be given the results of the systematic reviews and qualitative assessment in tabular form to review before providing their opinions.

Any evidence synthesis components employed in the process of questionnaire development will follow the relevant recommendations of the Institute of Medicine's Committee on Standards for Systematic Reviews of Comparative Effectiveness Research, especially regarding the qualitative assessment.[21]

We anticipate there will be nearly identical opinions for some KQs across the three sources of information. This will lead to significant redundancy which will unnecessarily burden the participants with repeating ratings on nearly identical opinions. For this reason, two independent assessors will thematically group the identical opinions and meet for consensus. After consensus on the groupings is achieved, they will then draft a single summary opinion. This pool of non-redundant opinions will be grouped together under their respective KQs and prepared for circulation to all participants in the form of an online questionnaire hosted on REDCap. The sources of the opinions will be masked to all participants to eliminate bias (eg, allegiance to a treatment approach).

### Phase 2: rating evaluation

Participants will be emailed the questionnaire and asked to rate each opinion individually on a 5-point Likert scale of agreement: 1=strongly disagree, 2=disagree, 3=neutral, 4=agree and 5=strongly agree. The questionnaire responses will be deidentified and the opinions will be grouped into their respective KQs. Once all the questionnaires have been received, the ratings will be statistically summarised for each opinion. This will involve calculating and graphically depicting the median, IQR and the percentages in each response category for each opinion, separately for patient/family/support persons and clinician/researcher groups. This statistical summary will be given as feedback to all participants, along with their own

rating, so they can see where their ratings are relative to others for each opinion under each KQ.

### Phase 3: rerating

Participants will be emailed the same questionnaire and asked to rerate their agreement on each opinion. This gives participants the opportunity to change their ratings in light of the feedback. After all the questionnaires have been received, the ratings will be statistically analysed using predefined consensus thresholds (table 2).

Opinions meeting threshold for essential, recommended or no consensus will be brought forward to the consensus meeting for review. If at least one opinion meets one of these thresholds, then we will organise a virtual video meeting. Opinions meeting threshold for exclusion will not be brought to the consensus meeting. If no opinions can be brought to the consensus meeting, then a third round of ratings will be performed. If no opinions can be brought forward after a third round, then the consensus meeting will be held on all the opinions.

### Phase 4: moderated consensus meeting

The purpose of this meeting is to finalise the set of opinions and phrasing for each KQ. We will have an independent chair for this meeting to minimise the disadvantages of group discussion by moderating the discussion. The chair will also help identify when significant differences of opinion between patient/family/support persons versus clinicians/researchers exist. Multiple meetings may take place at the discretion of the chair. The meetings will be recorded for reference if necessary and securely stored. However, all opinions will be considered final at the end of this consensus meeting process. Email correspondence will be allowed during phase 4 among the chair and participants in order to edit phrasing of opinions.

### Guideline development

The final opinions from phase 4 will be the recommendations in the guideline, which will be written up by the GWG. No changes to the opinions will be made by the GWG. If there are significant differences of opinion, the guideline will present the differing recommendations alongside one another.

### Conflict of interest management

All members of the GWG and potential participants will submit a conflict of interest (COI) disclosure form,

| Table 2 | Consensus thresholds |
|---|---|
| Essential | The first quartile (Q1) of the ratings for the opinion is >3 (neutral) in both the patient/family/support persons and clinician/researcher groups. |
| Recommended | The first quartile (Q1) of the ratings for the opinion is >3 (neutral) in either the patient/family/support persons or clinician/researcher groups. |
| Exclusion | The third quartile (Q3) of the ratings for the opinion is <3 (neutral) in both the patient/family/support persons and clinician/researcher groups. |
| No consensus | Essential, recommended or exclusion thresholds have not been met. |

developed based on the International Committee of Medical Journal Editors form. Potential participants will have to submit this form along with the demographic form in order to be considered for eligibility to participate in the expert panel. COIs will be considered by the study coordinator and, if needed, by the GWG, to determine eligibility. The principle guiding eligibility decisions is to ensure that COIs represent no more than a minority of all study participants. The chair of the consensus meeting will have no COI. Any reported COI will be discussed by the GWG and documented in the guideline in light of its potential impact on the proposed recommendations. This COI data will be collected before consent is obtained to participate in this study. Therefore, if a person is ineligible to participate or does not consent to participate in the study after reviewing the consent form, the COI form will be promptly deleted and will not be kept part of the study records.

## Statistical analysis

We will analyse the ratings using medians, IQRs and percentages for each response category. These values will also be graphically depicted in the feedback provided to the participants. We will statistically compare the agreement ratings between patient/family/support persons versus clinician/researcher groups using independent samples t-tests or Wilcoxon rank-sum tests depending on if parametric assumptions are met or not, respectively, and document these differences in the final version of the guideline. All tests of significance will be two tailed with $\alpha$ set at 0.05.

## ETHICS

This study has received ethics approval by the Hamilton Integrated Research Ethics Board in Hamilton, Ontario, Canada.

## DISSEMINATION PLAN

We anticipate that the guideline emerging from this Delphi study will be published in a high-impact peer-reviewed journal. The guideline will be disseminated to regional, national and international audiences through presentations at regional, national and international conferences. Finally, we will disseminate the guideline through the national and international associations for psychiatry, psychology, emergency medicine, nursing and social work.

## DISCUSSION

Clinicians caring for adults with BPD in acute settings such as the ED have little guidance or evidence to base their decisions on. In this vacuum, patients are at risk of a wide variety of treatment approaches and practices, ranging from excellent to iatrogenic. Therefore, specific and detailed guidance for managing adults with BPD in

the ED is urgently needed for the sake of patients and their families/support persons. Consensus methods such as Delphi play an important role in science where there is limited or contradictory evidence.[16] Through a structured process, these methods assess the degree of agreement and/or resolve disagreement among experts. The consensus can provide helpful scaffolding for clinical guidelines until empirical study fills the gaps. Furthermore, the clinical guidelines will provide important hypotheses for these empirical investigations. This is the rationale for our Delphi study. The opinions emerging from this study will become the first consensus guideline on the management of adults with BPD in the ED.

The nominal group technique is another consensus method used in health research.[15] It uses a facilitated meeting of experts who gather information, rate, discuss and rerate/discuss a series of items, questions or opinions. Each participant contributes their ideas to the facilitator who thematically groups the ideas for discussion. The ideas are then rated, summarised and the summary is fed back to the participants for discussion and rerating. The final ratings are then summarised and evaluated for consensus using predefined thresholds. The disadvantage of this technique is that it is often not feasible for all relevant experts to participate in the meeting. This is why typically the nominal group technique usually involves small groups of 9–12 individuals.[16] Moreover, experts often do not have the time to devote to these steps commensurate with the complexity of the issues. By contrast, the Delphi method allows researchers to synthesise a large amount of information across many individuals with relatively minimal imposition on the participants' time.

The main limitation of this study is that, even if we achieve consensus on some or all opinions, this does not guarantee that the correct recommendations have been found. Consensus does not necessarily imply truth. Only empirical studies can validate the opinions generated from a consensus process, which at this time are lacking. Therefore, our study is best viewed as a structured sampling of expert opinions rather than an attempt to reach a definitive conclusion about what are truly the best practices in managing adults with BPD in the ED. Another limitation is that our expert panel will not be representative of the disciplines, backgrounds and treatment approaches we are sampling. Relatedly, we anticipate there will be more clinician/researcher participants on the panel, which risks silencing patient/family/support persons voices. This risk will be mitigated by documenting and discussing the recommendations of patient/family/support persons if significant differences of opinion exist. Additionally, this study does not focus on guidelines for paediatric patients with BPD. While many of the guidelines here will be relevant for paediatric patients, a logical next step would be to develop guidelines for that population, given the inherent differences in the assessment and management of adults versus youth. Notwithstanding these limitations, we hope the guidance emerging from this Delphi study

will be a helpful clinical resource to improve the care of adults with BPD in the ED.

**Author affiliations**
[1]Department of Psychiatry and Behavioural Neurosciences, McMaster University, Hamilton, Ontario, Canada
[2]Department of Psychiatry, University of Michigan, Ann Arbor, Michigan, USA
[3]Institute of Psychology, University of Wroclaw, Wroclaw, Poland
[4]Meta-Research Centre, University of Wroclaw, Wroclaw, Poland
[5]McMaster Integrative Neuroscience Discovery and Study (MiNDS), McMaster University, Hamilton, Ontario, Canada
[6]Mood Disorders Program, St. Joseph's Healthcare, Hamilton, Ontario, Canada

**Acknowledgements** We would like to express our gratitude to the people with BPD and their family members/support persons for reviewing and commenting on an earlier draft of this protocol, as well as providing advice on the key questions for the guideline. We would also like to thank the patients with BPD who we have cared for and learned from over the years. Their lives inspired this project.

**Contributors** AP, VH, BH, DF, JP, PR, BNF and PL are responsible for the design of the study and early drafts of the protocol. PL is the principal investigator. AP is the study coordinator and wrote the first draft of the protocol. AP, VH, BH and PL are responsible for final draft of the protocol manuscript. All authors have read and approved the final manuscript, provided final approval for publication, and agree to be accountable for all aspects of this work. AP attests all listed authors meet authorship criteria and that no others meeting criteria have been omitted.

**Funding** This research was funded in part by National Science Centre, Poland (grant number 2021/41/B/HS6/02844). For the purpose of Open Access, the author has applied a CC-BY public copyright licence to any Author Accepted Manuscript (AAM) version arising from this submission.

**Competing interests** PL is a codeveloper of GPM and VH published an expert opinion on the management of BPD in the ED based on GPM principles.

**Patient and public involvement** Patients and/or the public were involved in the design, or conduct, or reporting, or dissemination plans of this research. Refer to the Methods section for further details.

**Patient consent for publication** Not applicable.

**Provenance and peer review** Not commissioned; externally peer reviewed.

**ORCID iDs**
Aaron Prosser http://orcid.org/0000-0002-9379-8378
Bartosz Helfer http://orcid.org/0000-0001-9863-2871
Benicio N Frey http://orcid.org/0000-0001-8267-943X

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
