## [Reviewer comments · BMJ Open]

ARTICLE DETAILS

TITLE (PROVISIONAL)	Protocol for a Delphi consensus study to develop guidelines for the management of adults with borderline personality disorder in the emergency department
AUTHORS	Prosser, Aaron; Hong, Victor; Helfer, Bartosz; Fudge, David; Patterson, Janet; Rosebush, Patricia; Frey, Benicio; Links, Paul

VERSION 1 – REVIEW

REVIEWER	Guilé, Jean Marc CHU Amiens-Picardie
REVIEW RETURNED	24-Jul-2023

GENERAL COMMENTS	DELPHI BPD The study is a relevant endeavor for this highly demanding population. It will adequately address the three targets: patient, staff and system. I would add the relatives/support network which is of major importance for the therapeutic alliance, motivation and outcome in BPD patient. Including health system users within the informants involved in the DELPHI procedure, is innovative and promising. Two main challenges/issues need clarification: - the age scope of the review since the involved support and health systems are different in youth and adults. There is no information (item 5 in the appendix). Including youths is highly relevant since BPD adolescents are high ER consumers.- comorbidity. According to the meta-analyses and clinical practices, co-occurrence of disorders such as substance use, or depressive disorders is of pivotal impact on the choice of intervention and the resulting efficacy. With respect to the DELPHI process, a few issues should be clarified: -how the members of the expert panel will be chosen? How many, and which geographical/country representativeness?-how many rounds of back and forth between experts and rewriting the questions are planned (between 2 and 4 according to the standards)? Comments on the RQ, KQ and outcomes RQ#1 while the review is focusing on intervention, should it be worth including a section on screening/assessment since it might be necessary to ascertain the diagnosis in ER, especially in youngster attending ER for the first time? RQ#1 and 2 should include comorbidity issues (substance use, depression) RQ#3 hospitalization means full time admission. What about part time hospitalization? Would it be included in RQ#4?
---

	RQ#6 it's mandatory to clarify whether the review encompasses youth, since the social support network includes parents and peers (which is not included in key words) and health trajectories different from adult' ones. KQ#1 and 4 why is there no keywords relative to medication and psychotherapy as well as community or internet-based program? KQ#9 employment and school (depending on the age range of the review) should be included. O#1 with respect to the high comorbidity between suicidal behaviors and NSSI, the later should be included. O#4 why not including psychotherapy and OPD key words? Minor comments References should be checked: e.g., NICE, APA and NHMRC are rather 5,6,7. Ref #37 should be rewriting = Paris J.
--	---

REVIEWER	di Giacomo, Ester University of Milano-Bicocca
REVIEW RETURNED	Well done, congratulations!

VERSION 1 – AUTHOR RESPONSE

Reviewer: 1

Dr. Jean Marc Guilé, CHU Amiens-Picardie

1. The study is a relevant endeavor for this highly demanding population. It will adequately address the three targets: patient, staff and system. I would add the relatives/support network which is of major importance for the therapeutic alliance, motivation and outcome in BPD patient. Including health system users within the informants involved in the DELPHI procedure, is innovative and promising.

Response: Thank you for the positive feedback.

2. Two main challenges/issues need clarification: - the age scope of the review since the involved support and health systems are different in youth and adults. There is no information (item 5 in the appendix). Including youths is highly relevant since BPD adolescents are high ER consumers.

Response: Thank you for bringing this important consideration forward. We agree that including youth with BPD is highly relevant to this topic. There is an urgent need for a child/adolescent guideline on the management of youth with BPD who present to the emergency department in mental health crises. The conception and development of this project had adults with BPD in mind given that the collective expertise of the Guideline Working Group (GWG) is in the care of adults. As a result, there is a tangible risk of the guidelines emerging from this Delphi study will not adequately capture the complexities and needs of caring for youth with BPD. For this reason, we have made it more clear in the protocol and study design that our focus is on adults with BPD. That being said, we anticipate that many of the principles and guidance that will be discussed will be relevant to child/adolescent mental health services because of the continuity of psychopathology between youth and adults with BPD. However, some may not transfer because of the differences between adult vs. child/adolescent mental health services. This includes differences in service provision, involvement of families, school systems, social services, and community agencies. For these reasons, rather than combining adults and youth in a single guideline, we would encourage the development of a specific guideline for youth with BPD. This would be the logical next task to undertake after the completion of our Delphi study.

3. comorbidity. According to the meta-analyses and clinical practices, co-occurrence of disorders such as substance use, or depressive disorders is of pivotal impact on the choice of intervention and the resulting efficacy.

Response: We appreciate both the prevalence and importance of comorbidities when managing individuals with borderline personality disorder. That being said, treatment decisions that are impacted by the presence or absence of comorbidities are typically of greater concern in a longer term setting as opposed to the acute emergency setting, where the focus is more on managing acute distress and doing a risk assessment. However we have made an addition to the introduction and amendments to KQ1 and KQ2 to address this concern.

4. With respect to the DELPHI process, a few issues should be clarified:-how the members of the expert panel will be chosen? How many, and which geographical/country representativeness?

Response: There are many demographic variables to consider when balancing the expert panel to ensure representativeness. In line with current approaches to equity, diversity, and inclusion in research, we have added a section to the protocol on representation (page 9). We will try to build a balanced panel of experts based on the following demographic data: discipline (psychiatry, psychology, nursing, emergency medicine, social work), group (clinician, researcher, person with BPD, family member/support person), treatment allegiance, ethnicity, gender, and sexual orientation. We have developed a basic demographic data screening form which will be emailed to potential participants who expressed interest to participate in the study. We cannot know in advance the demographics and relative percentages of our pool of potential participants who express interest to participate in our study. Therefore, saturation will have to be determined on a case-by-case basis amongst the GWG. We have decided not to include geography/country representativeness because our main concern for equity, diversity, and inclusion is in relation to the above demographic factors.

5. -how many rounds of back and forth between experts and rewriting the questions are planned (between 2 and 4 according to the standards)?

Response: I believe you are referencing Phase 4 of the study. We have decided to not define a priori the number of rounds for editing the final phrasing of the opinions amongst the expert panel. Our rationale is because we cannot know with certainty in advance the nature and level of disagreement between the panelists on the opinions for each KQ. If there is a significant amount of disagreement, it may take time for the panel to converge on a consensus about the final phrasing. If we define this a priori in the protocol, we risk artificially limiting the time for deliberation which may reduce the quality of the final guideline.

6. Comments on the RQ, KQ and outcomes RQ#1 while the review is focusing on intervention, should it be worth including a section on screening/assessment since it might be necessary to ascertain the diagnosis in ER, especially in youngster attending ER for the first time?

Response: Because early recognition and diagnosis of BPD is crucial for appropriate interventions, we feel it is reasonable to add a key question to address this issue. We've added this as KQ3 and adjusted the numbers of the other KQs accordingly.

7. RQ#1 and 2 should include comorbidity issues (substance use, depression)

Response: As noted above, we have now included this consideration.

8. RQ#3 hospitalization means full time admission. What about part time hospitalization? Would it be included in RQ#4?

Response: Hospitalization is referring to a psychiatric admission. Partial hospital programs would be considered an aftercare service, which is captured under KQ5.

9. RQ#6 it's mandatory to clarify whether the review encompasses youth, since the social support network includes parents and peers (which is not included in key words) and health trajectories different from adult' ones.

Response: This Delphi study and guidelines emerging from it will pertain only to adults with BPD.

10. KQ#1 and 4 why is there no keywords relative to medication and psychotherapy as well as community or internet-based program?

Response: In the body of the protocol on page 6, we explained that KQ is regarding psychosocial and pharmacological interventions. We've now made this explicit in the wording of KQ1. Regarding KQ5 (previously KQ4), we mean psychosocial aftercare services, which would include community and internet-based programs. We are excluding pharmacotherapeutic interventions since that would essentially turn this KQ into a systematic review of the evidence for pharmacotherapy for outpatients with BPD, which is beyond the scope of this study. This has also been recently reviewed with the update of the 2010 Cochrane review: <https://pubmed.ncbi.nlm.nih.gov/32504127/>. We've updated the wording of the KQ to make sure this is more clear. The search terms of KQ1 are broad enough to include psychosocial and pharmacological interventions, however the latter would be restricted to pharmacological interventions specifically for patients with BPD in the emergency department, not outpatients.

11. KQ#9 employment and school (depending on the age range of the review) should be included.

Response: KQ10 (previously KQ9) is not concerned with environment in the sense that we believe you are referring to. Rather, it refers to the physical layout and interior design of the emergency department.

12. O#1 with respect to the high comorbidity between suicidal behaviors and NSSI, the later should be included.

Response: This is already included in the suicidality outcomes (page 6 of the Supplementary Materials).

13. O#4 why not including psychotherapy and OPD key words?

Response: The reason is because these outcomes are concerned with service utilization in terms of the broad categories of care (e.g., hospitalization, attending outpatient appointments) rather than the specific sub-categories (e.g., psychotherapy vs. medication management).

14. Minor comments References should be checked: e.g., NICE, APA and NHMRC are rather 5,6,7. Ref #37 should be rewriting = Paris J.

Response: Thank you for noticing this. We've corrected this.

Reviewer: 2

Dr. Ester di Giacomo, University of Milano-Bicocca

1. Well done, congratulations!

Response: Thank you for the positive feedback!